# Experimental study of high-flow and low-expansion backfill material

Cheng Wang[1,2], Chun Wang[1,2]*, Zuqiang Xiong[2], Yuli Wang[3‡], Yafeng Han[4‡]

1 School of Energy Science and Engineering, Henan Polytechnic University, Jiaozuo, China, 2 Collaborative Innovation Center of Coal Work Safety of Henan Province, Jiaozuo, Henan, China, 3 School of Materials Science and Engineering, Henan Polytechnic University, Jiaozuo, China, 4 School of Civil Engineering, Chongqing University, Chongqing, China

☉ These authors contributed equally to this work.
‡ These authors also contributed equally to this work.
* 657156842@qq.com

**Data Availability Statement:** Data is available from Dryad: https://doi.org/10.5061/dryad.hmgqnk9d8.

**Funding:** The organizations that supported our study are listed as follows: CXTD2017088 (Zuqiang Xiong), 201102310217 (Cheng Wang),

## Abstract

High-flow low-expansion backfill materials have been developed to improve difficult slurry pipeline transport and poor roof-contact effect of many filling materials. The fly ash content was fixed at 80%, with 8.5% - 9.5% mineral powder content, 8.5% - 9.5% lime, 2% - 3% desulfurized gypsum, 0.9% - 1.2% sodium carbonate, and 0.01% - 0.02% aluminum powder content. The prepared backfill material processed good fluidity, with the expansion rate of the hardened material reaching 2% - 3%, and compressive strength on 90 d reaching 4 MPa—5.5 MPa. SEM observations indicated that as the aluminum content increased, ettringite on bubble walls transformed from a fine-needle to needle-rod shape. Secondly, the hydration products of the system were mainly hydrated calcium silicate gel and ettringite, which interconnected and promoted the formation of the structure. The backfill material has extensive sources of raw materials, low cost, simple filling process, and good filling effect.

## Introduction

Coal is the main energy source in China and responsible for a considerable portion of energy consumption, hence, it is important to realize a low-cost method to exploit coal resources with high utilization and low emission rates. Backfill mining is an important component of green mining and has the potential to increase the recovery of coal resources, control the overlying strata movement, and maximize the recycling of solid waste. The utilization of backfill materials is the main approach to realize these goals. For example, the total amount of coal compression in the Wangtaipu coal mine is as high as 57.46 million tons. At this level of resource depletion, it is vital to focus research on backfill mining processes to extract "three-low" pressed coal. During backfill mining, the core is used as backfill material, which not only determines the backfill process, but also contributes its cost. Therefore, new mining engineering materials will play a crucial role in the development of green mining [1–3]. As determined by commonly-used backfill materials, the backfill processes in China include waste rock dry filling, sand water filling, tail filling, cemented filling with unclassified tailings, and paste and paste-like filling [4–9]. These processes are complex and have large transportation volumes,

192102310247(Cheng Wang) and 20A440007 (Cheng Wang), are supported by Henan Polytechnic University, Henan province, 51904093 (Chun Wang) is supported by National natural science foundation of China.

**Competing interests:** NO authors have competing interests.

high labor costs, and non-ideal goaf densities. In order to overcome these disadvantages, extensive research has been carried out on high-flow backfill materials. For example, the ultra-high water filling material developed by Prof. Feng Guangming exhibits high strength when prepared at relatively high water-to-cement ratios [10, 11]. However, the deficiencies of using a single material source and low backfill strength of ultra-high water filling materials have limited the broader applications of these materials. In order to further simplify the filling process, expand material sources, and increase the tight-filling ratio, a high-flow low-expansion backfill material has been developed from fly ash in this study.

## Materials and methods

### Materials

The fly ash has the potential pozzolanic activity. The wide source and low price of the fly ash are also its advantage. Thus, the fly ash was selected as the main component of the backfill materials. Commercially available fly ash was supplied by Shanxi Yang City Power Plant and used as the main backfill material. It had a density of 2.21 g/cm$^3$, a specific surface area of 332.1 m$^2$/kg, and a water content of 0.05%. The chemical composition of the fly ash is listed in Table 1. The mineral powder, lime, and desulfurized gypsum were selected as the cementitious material. Mineral powder was supplied by Jinchneg City Fusheng Steel Plant, with a specific surface area of 427.8 m$^2$/kg and density of 2.65 g/cm$^3$. The lime was produced locally in Jincheng City, with an effective calcium oxide content of 73%. The sieve residue from an 80 μm square mesh sieve was 15.6%. Desulfurized gypsum was obtained from a local power plant, which had a main component of $CaSO_4 \cdot 2H_2O$ and water content of 16.56%. The sodium carbonate was used as the alkali-activator to improve the hydration activity of the fly ash. Sodium carbonate was industrial-grade with purity above 98%. The aluminum powder is used as air-entraining agent. Aluminum powder was water formulation aluminum powder paste in the building material industry standard JC/T 407–2008 "Aluminum Powder Paste for Aerated Concrete." Purified tap water was used for all experiments.

### Testing methods

The slump flow test of the slurry was performed according to Appendix A in GB 50119–2003. The dimension of the truncated cone shape mode was: 60±0.5 mm in height, 36±0.5 mm in upper opening inner diameter, and 60±0.5 mm in lower opening inner diameter. The specimen was molded using a testing mold 70.7 mm × 70.7 mm × 70.7 mm in size (GB/T 50080–2016). For maintenance, the relative humidity was above 90%, the temperature was 20±1˚C, and compressive strength was tested on the 7[th], 28[th], and 90[th] day. The expansion rate [12] of the consolidated slurry was defined as the ratio of the interface height differences before and after slurry hardening to the interface height value of the initial slurry. The determination thought of the expansion rate is consistent with the expansive ratio determination of expansive cement (JC/T 313–2009). The SEM sample was taken from the specimens in uniaxial compression test, the sample was made by 1cm*1cm. Then it was fixed on the metal plate with double-sided conductive adhesive, the metal plate was placed in a gold-spraying apparatus for gold spraying, drying and vacuuming, and finally placed in a SEM instrument for the experiment.

**Table 1. The main chemical composition of raw materials.**

| Sample | SiO$_2$ | Al$_2$O$_3$ | Fe$_2$O$_3$ | CaO | MgO | Na$_2$O | K$_2$O | SO$_3$ | TiO$_2$ | loss in ignition |
|---|---|---|---|---|---|---|---|---|---|---|
| **Fly ash** | 47.18 | 26.75 | 10.08 | 3.47 | 2.54 | 1.21 | 1.85 | 0.76 | 0.7 | 5.16 |
| **Mineral powder** | 32.72 | 15.62 | 0.9 | 37.6 | 8.97 | 0.32 | 0.47 | 0.31 | 0.65 | 0.94 |

## Results and analysis

The performance of the backfill materials is balanced when the mass ratio of the fly ash is 80% based on the initial investigation. The strength of the backfill material is sufficient at this ratio. Additionally, the cost of the backfill material is also cheap. Thus, the mass ratio of the fly ash was fixed at 80% in this study. A series of experiments were designed to determine the amount of the cementitious material, the alkali-activator, and the air-entraining agent. During tests, the ratio of each component was changed, and the expansion rate, fluidity, and compressive strength were used to determine the optimum ratio among various components.

### Determination of mineral powder and lime ratio

The ratio of water-to-material was fixed at 0.7, with the percentages of fly ash, sodium carbonate, and aluminum powder being 80%, 1%, and 0.02%, respectively. When the mineral powder -to-lime content was fixed at 20%, the change in ratio allowed for comparison of ratio influence on the slump flow, expansion rate, and compressive strength. The detailed ratios and results are listed in Table 2.

As shown in Table 2, the mineral powder amount decreases with increasing lime amount, resulting in the slump flow of the backfill material to gradually decrease. Fluidity improves as the mineral powder amount increases with decreasing lime ratio. As the amount of mineral powder decreases, the expansion rate of the backfill material first increases and then decreases. When the amount of mineral powder is 5% - 20%, the expansion rate of the backfill material is 2.5% - 3.5%. The compressive strength first increases and then decreases with increasing mineral powder amount from 0% to 15% and decreasing lime ratio.

The above analyses show that at 5% - 15% mineral powder amount and 15% - 5% lime amount, the fluidity, expansion rate, and compressive strength are relatively good. When the amount of mineral powder and lime is fixed at 20%, 5% - 15% mineral powder shows relatively good performance.

### Determination of desulfurized gypsum amount

Gypsum is a good activator of alkali-activated coagulation materials. The main components of desulfurized gypsum are sulfates and crystal water, which can play inhibitory roles in the digestion of lime [13, 14], further affecting the material performance. In order to quantitatively analyze the influence of desulfurized gypsum on material performance, local desulfurized gypsum was used for comparison. In the tests, the water-to-material ratio was fixed at 0.7, and the combined content of mineral powder, lime, and desulfurized gypsum was fixed at 20%. Desulfurized gypsum was used to replace an equal proportion of mineral powder and lime, and the

**Table 2. Influence of amounts of mineral powder and lime on material performance.**

| No. | The mass of raw material /% | | | | | ED /mm | ER /% | Compressive /MPa | |
|---|---|---|---|---|---|---|---|---|---|
| | FA | MP | QL | SC | AP | | | 7 d | 28 d |
| 1 | | 20 | 0 | | | 275 | 1.8 | — | 0.14 |
| 2 | | 15 | 5 | | | 230 | 2.6 | 0.87 | 1.26 |
| 3 | 80 | 10 | 10 | 1 | 0.02 | 205 | 3.43 | 1.04 | 1.89 |
| 4 | | 5 | 15 | | | 195 | 2.89 | 0.78 | 2.39 |
| 5 | | 0 | 25 | | | 165 | -1.92 | 0.46 | 1.99 |

FA is fly ash; MP is mineral powder; QL is quick lime; DG is desulfurized gypsum; SC is sodium carbonate; AP is aluminum powder; W/C is water cement ratio; ED is extension degree; ER is expansion rate.

**Table 3. Influence of desulfurized gypsum amount on material performance.**

| No. | The mass of raw material /% | | | | | | ED /mm | ER /% | Compressive /MPa | |
|---|---|---|---|---|---|---|---|---|---|---|
| | FA | MP | QL | DG | SC | AP | | | 7 d | 28 d |
| 1 | | 10.0 | 10.0 | 0 | | | 210 | 3.39 | 1.01 | 1.91 |
| 2 | | 9.5 | 9.5 | 1 | | | 220 | 3.21 | 0.9 | 2.32 |
| 3 | 80 | 9.0 | 9.0 | 2 | 1 | 0.02 | 235 | 3.1 | 1.09 | 3.89 |
| 4 | | 8.5 | 8.5 | 3 | | | 231 | 2.14 | 1.12 | 3.92 |
| 5 | | 8.0 | 8.0 | 4 | | | 235 | 1.86 | 1.24 | 4.21 |

influence of changing the ratio among the three materials on the backfill material performance was compared. The testing ratios and results are shown in Table 3.

Table 3 shows that the slump flow of the backfill slurry gradually increases with increasing desulfurized gypsum ratio. When the amount is 2%, the slump flow remains almost stable. However, as the lime amount increases, the expansion rate of the slurry gradually decreases. At 1% - 3% desulfurized gypsum amount, the expansion rate of slurry remains within 3.21% - 2.14%. The compressive strength gradually increases with desulfurized gypsum content, which is greater than the un-doped material. For amounts between 2% - 4%, the strength on 28 d reaches 3.89–4.21 MPa. By combining the influence of desulfurized gypsum on the performances of the three materials, the optimal amount is 2% - 3%.

## Determination of activator amount

Sodium carbonate was used as the alkali-activator during tests. The amount of sodium carbonate was controlled within 0% - 1.2% and the amounts of fly ash and cementing material remained constant at 80% and 20%, respectively. The water-to-material ratio of the slurry was fixed at 0.7. The expansion rate, slump flow, and strength of specimens at different ages were observed, and the testing ratios and results are listed in Table 4.

Table 4 shows that as the activator amount increases, the slump flow of the slurry gradually decreases, and expansion rate and compressive strength at different ages gradually increase. At 0.9% - 1.2% of activator amount, the backfill material expansion rate remains within the ideal range of 2.76% - 3.09%, and compressive strength on 28 d is within 3.63–4.2 MPa. The optimal amount of sodium carbonate is 0.9% - 1.2%, based on its influence on the three indices.

## Determination of aluminum powder ratio

To ensure a good top-filling effect, the expansion characteristics of the slurry were explored. The water-to-material ratio was fixed at 0.7, and the amounts of fly ash and cementing material

**Table 4. Influence of sodium carbonate amount on material performance.**

| No. | The mass of raw material /% | | | | | | ED /mm | ER /% | Compressive /MPa | |
|---|---|---|---|---|---|---|---|---|---|---|
| | FA | MP | QL | DG | SC | AP | | | 7 d | 28 d |
| 1 | | | | | 0 | | 275 | 0.87 | 0.87 | 1.91 |
| 2 | | | | | 0.3 | | 262 | 1.26 | 1.26 | 2.18 |
| 3 | 80 | 9 | 9 | 2 | 0.6 | 0.02 | 240 | 1.47 | 1.47 | 2.53 |
| 4 | | | | | 0.9 | | 232 | 2.76 | 2.76 | 3.63 |
| 5 | | | | | 1.2 | | 220 | 3.09 | 3.09 | 4.2 |

Sodium carbonate was added at the mass percentage of the total material amount and added as per the method.

**Table 5. Influence of aluminum amount on the material performance.**

| No. | The mass of raw material /% | | | | | | ED /mm | ER /% | Compressive/MPa | |
|---|---|---|---|---|---|---|---|---|---|---|
| | FA | MP | QL | DG | SC | AP | | | 7 d | 28 d |
| 1 | | | | | | 0 | 231 | 0 | 2.04 | 4.5 |
| 2 | | | | | | 0.01 | 240 | 1.02 | 1.97 | 4.23 |
| 3 | 80 | 9 | 9 | 2 | 1 | 0.02 | 235 | 3.1 | 1.53 | 3.89 |
| 4 | | | | | | 0.03 | 248 | 5.41 | 0.97 | 2.4 |
| 5 | | | | | | 0.04 | 240 | 7.2 | 0.87 | 1.9 |

Aluminum powder was added at the mass percentage of total material amount and added as via the described method.

were kept at 80% and 20%, respectively. The sodium carbonate amount was set to 1%, and aluminum powder amount varied between 0.01% - 0.04%. The test ratios and results are shown in Table 5.

Table 5 shows that by changing aluminum powder amount, no obvious influence on the expansion rate of slurry is observed, which remains within 240±10 mm. As the amount of aluminum powder increases, the slurry expansion rate obviously increases. However, between 0.01% - 0.02%, the slurry expansion rate remains within 1.02% - 3.1%. As the aluminum powder amount increases, the slurry expansion rate exhibits high tunability. By increasing the expansion rate, the strength of the specimens at different ages gradually decreases. At 0.01% - 0.02% aluminum amount, the compressive strength of the specimen on 28 d remains in the ideal range of 3.89–4.23 MPa. Therefore, the optimum aluminum amount is 0.01% - 0.02%.

### Backfill material characteristics tests under different water-to-material ratios

In the tests, the fly ash amount was fixed at 80%, while those of mineral powder, lime, and desulfurized gypsum were 9%, 9%, and 2%, respectively. 1% sodium carbonate and 0.02% aluminum powder were used, and water-to-material ratio was controlled within 0.55–0.75. The slurry slump flow, expansion rate, strength, and consumption per cubic dry material were examined. The experimental ratios and results are shown in Table 6.

Table 6 shows that the water-to-material ratio significantly influences the slurry expansion rate and compressive strength. As water content increases, the fluidity of the slurry increases, while the compressive strength of specimens obviously decreases. By increasing water-to-material ratio, a slight change in backfill material expansion rate is observed, and remains in the ideal range of 2% - 3%.

Combined with Fig 1, this shows that the low water-to-material ratio obviously improves the compressive strength of specimens at different ages. At 0.6–0.7 water-to-material ratio, the

**Table 6. Influence of the water-to-material ratio on the material performance.**

| No. | The mass of raw material /% | | | | | | W/C | ED/mm | ER/% | Consumption of dry material | Compressive /MPa | | |
|---|---|---|---|---|---|---|---|---|---|---|---|---|---|
| | A | MP | QL | DG | SC | AP | | | | /t/m³ | 7 d | 28 d | 90 d |
| 1 | | | | | | | 0.55 | 163 | 2.41 | 0.832 | 2.43 | 5.3 | 6.07 |
| 2 | | | | | | | 0.60 | 178 | 2.52 | 0.758 | 2.07 | 4.62 | 5.54 |
| 3 | 80 | 9 | 9 | 2 | 1 | 0.02 | 0.65 | 225 | 2.96 | 0.756 | 1.94 | 4.11 | 5.21 |
| 4 | | | | | | | 0.70 | 235 | 3.1 | 0.718 | 1.53 | 3.89 | 4.32 |
| 5 | | | | | | | 0.75 | 255 | 2.15 | 0.68 | 0.92 | 2.8 | 3.23 |

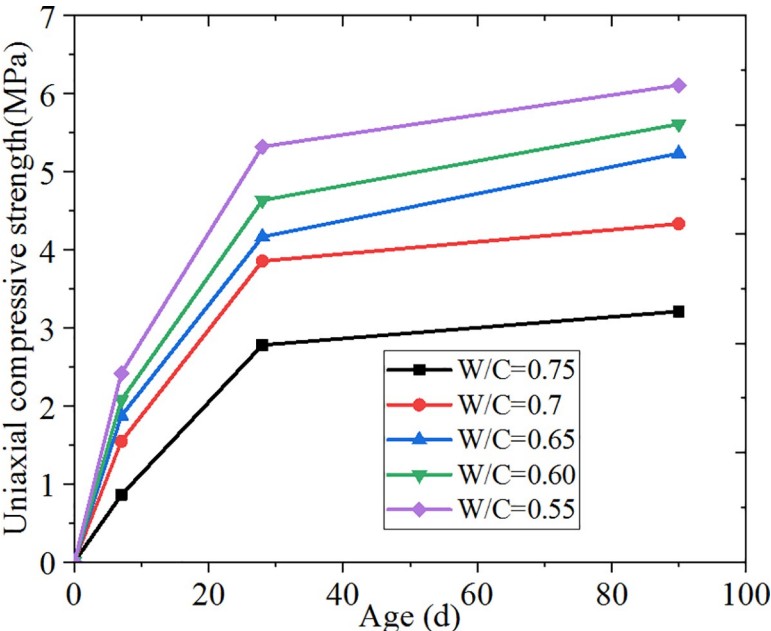

**Fig 1. Strength of specimens with different water-to-material ratios at various ages.**

strength of the backfill on 90 d is 5.5–4 MPa. The expansion rate remains within 178 mm—235 mm, with optimal cost. The calculated cost of the dry material is approximately 0.758–0.718 t/m$^3$, and backfill cost is 11.2 dollar /m$^3$.

## SEM analysis

The formation mechanism of the system structure was analyzed using SEM images. Fig 2 shows the hydration hardening structure with different aluminum amounts. As shown in Fig 2A-1, 2B-1 and 2C-1, the amount of aluminum powder increases, with increasing number of bubbles and bubble diameter. Due to the presence of calcium hydroxide, aluminum powder underwent the reaction shown in Eq (1), generating H$_2$ and hydrated calcium aluminate.

Due to the presence of gypsum, the reaction described in Eq (2) occurred in hydrated calcium aluminate, producing ettringite. To investigate the reason for ettringite formation, the influence of different aluminum powder amounts and bubbles in the structures was examined (Fig 2A-2, 2B-2 and 2C-2). As the aluminum powder content increases, ettringite around the bubbles gradually transforms from a fine needle-shaped to needle rod-shaped particles. This indicates that, at a constant desulfurized gypsum amount, as more hydrated calcium aluminate is formed, the amount of formed ettringite increases.

$$Al + Ca^{2+} + OH^- + H_2O \rightarrow CAH + H_2 \uparrow \tag{1}$$

$$CAH + C\bar{S}H_2 + Ca^{2+} + H_2O \rightarrow C_3A \bullet 3C\bar{S} \bullet H_{32} \tag{2}$$

By comparing Fig 3C-1 and 3C-2, at 0% desulfurized gypsum amount (Fig 3C-1), a large amount of network-like CSH (II) gel on the hardening structure surface and in the hydrated pores is observed. This gel ensures the strength of the slurry. However, since the structure of CSH (II) gel is dense, it covers the fly ash glass microsphere surface and forms a coating which inhibits further hydration reactions [15, 16]. Hydration products with high intensity are not

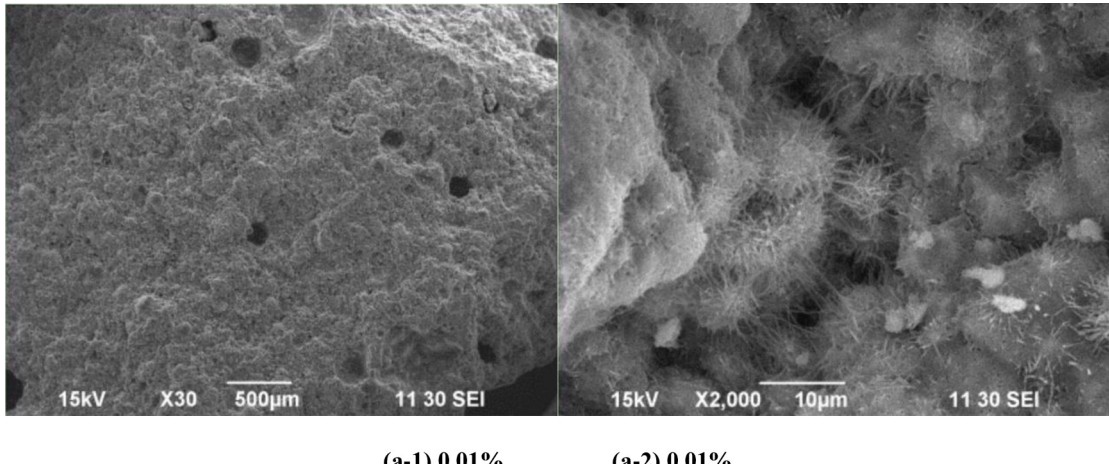

(a-1) 0.01%          (a-2) 0.01%

(b-1) 0.02%          (b-2) 0.02%

(c-1)0.03%          (c-2)0.03%

**Fig 2. Hardening structures with different aluminum powder amounts.** (a-1) 0.01% (a-2) 0.01% (b-1) 0.02% (b-2) 0.02% (c-1) 0.03% (c-2)0.03%.

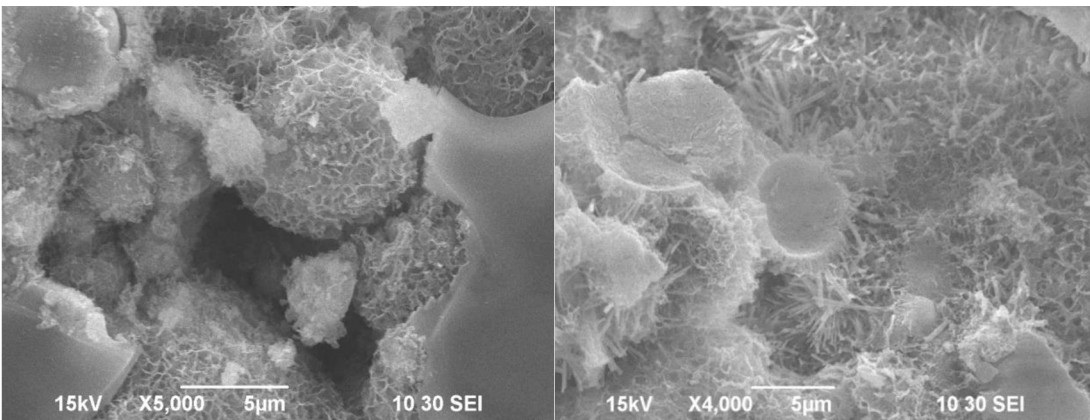

**(c-1) 0% desulfurized gypsum**          **(c-2) 2% desulfurized gypsum**

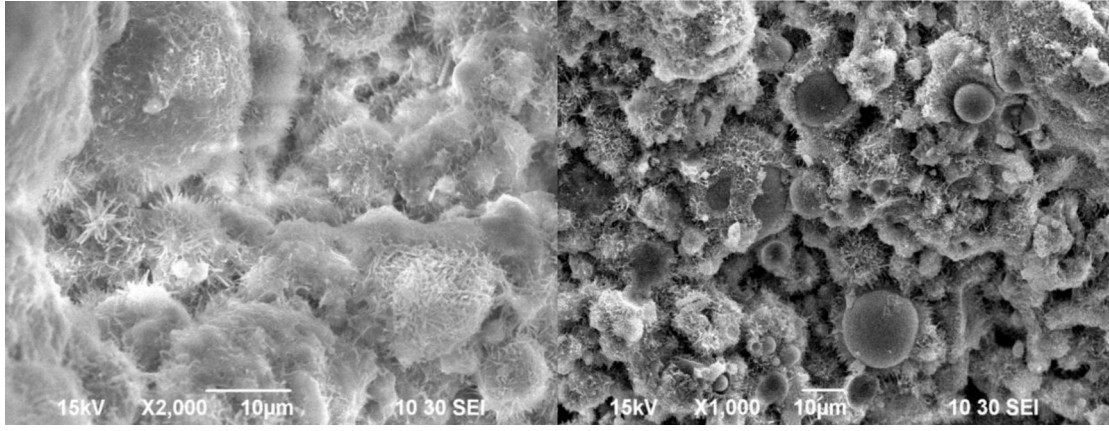

**(d-1) water-material ratio 0.6**          **(d-2) water-material ratio 0.7**

**Fig 3. Influence of desulfurized gypsum and water-to-material ratio on the hardening structure.** (c-1) 0% desulfurized gypsum (c-2) 2% desulfurized gypsum (d-1) water-material ratio 0.6 (d-2) water-material ratio 0.7.

continuously produced resulting in relatively low strength. At 2% desulfurized gypsum amount (Fig 3C-2), in addition to the network-like CSH (II) gel in the hydration system, obvious needle-rod shaped ettringite crystals are formed, promoting serious corrosion on the fly ash glass microsphere surface. As a sulfate activator [17, 18], the desulfurized gypsum largely improves the hydration extent of the fly ash active glass microspheres in the system, enriching the hydration products, and notably improving the strength compared to those in the absence of the sulfate activator. In addition, the comparison of Fig 3D-1 and 3D-2 shows that as the water-to-material ratio decreases, the gap between the spherical glass microspheres also decreases, and the density of the structure increases, as well as the strengths of specimens at different ages.

## Conclusions

When the fly ash amount was fixed at 80%, different amounts of mineral powder, lime, desulfurized gypsum, sodium carbonate, and aluminum powder were analyzed in order to

investigate their effects on the characteristics of backfill material. SEM was used to observe the microstructure of the obtained structures.

1. When the fly ash content was fixed at 80%, mineral powder was 8.5% - 9.5%, lime was 8.5% - 9.5%, desulfurized gypsum was 1% - 3%, with 0.9% - 1.2% sodium carbonate and 0.01–0.02% aluminum powder. When water-to-material ratio was controlled between 0.60–0.70, the backfill material showed good fluidity. The expansion rate of the hardened structure reached 2% - 3%, and the compressive strength of the sample on 90 d reached 4 MPa—5.5 MPa.

2. SEM images showed that as the amount of aluminum powder increased, ettringite around the bubbles transformed from a fine-needle shape to needle-rod shape. Then, as the amount of desulfurized gypsum increased, the hydration degree of the fly ash active glass micro-spheres in the slurry system also increased. Furthermore, low water-to-material ratio markedly increased the density of the hardening system, which improved the specimen strength at different ages.

3. The backfill material can be sourced from various low-cost raw materials, and also has a simple backfill process and effects.

## Acknowledgments

We thank Mr. Xiong from the Henan Polytechnic University for his expertise and kind assistance in this study.

## Author Contributions

**Data curation:** Yuli Wang.

**Investigation:** Zuqiang Xiong, Yafeng Han.

**Writing – original draft:** Cheng Wang.

**Writing – review & editing:** Chun Wang.

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
