## [Decision Letter · Decision Letter 0]

14 Apr 2020

PONE-D-20-00990

Experimental study of high-flow and low-expansion backfill material

PLOS ONE

Dear Mr. WANG,

Thank you for submitting your manuscript to PLOS ONE. After careful consideration, we feel that it has merit but does not fully meet PLOS ONE’s publication criteria as it currently stands. Therefore, we invite you to submit a revised version of the manuscript that addresses the points raised during the review process.

ACADEMIC EDITOR: In my point of view, this manuscript is  required additionally an explanation on the experimental design in the experimental part. Please explain the role of each material used clearly and explain the amount basis of each material used in the experiments to strengthen scientific method. 

We would appreciate receiving your revised manuscript by May 29 2020 11:59PM. To enhance the reproducibility of your results, we recommend that if applicable you deposit your laboratory protocols in protocols.io, where a protocol can be assigned its own identifier (DOI) such that it can be cited independently in the future. For instructions see: http://journals.plos.org/plosone/s/submission-guidelines#loc-laboratory-protocols

We look forward to receiving your revised manuscript.

Kind regards,

Kedsarin Pimraksa, PhD

Academic Editor

PLOS ONE

3. Thank you for stating the following financial disclosure: "no"

Reviewers' comments:

Reviewer's Responses to Questions

**Comments to the Author**

1. Is the manuscript technically sound, and do the data support the conclusions?

Reviewer #1: Yes

Reviewer #2: Yes

2. Has the statistical analysis been performed appropriately and rigorously? 

Reviewer #1: I Don't Know

Reviewer #2: Yes

3. Have the authors made all data underlying the findings in their manuscript fully available?

Reviewer #1: Yes

Reviewer #2: Yes

4. Is the manuscript presented in an intelligible fashion and written in standard English?

Reviewer #1: No

Reviewer #2: No

5. Review Comments to the Author

Reviewer #1: Table 1: Change loss by loss in ignition

Line 65: degree? ºC?

Line 158: changed Yuan by dollar?

Line 186: add gel at CSH (II)

The references are very local, they should be more international.

English must be revised

Reviewer #2: The author conducted an experimental study on high-flow and low-expansion backfill material, and the backfill material had extensive sources of raw materials, low cost, and a simple filling process and good filling effect. This is of great significance to the comprehensive utilization of mine solid waste and environmental protection. Nevertheless, I still make a few suggestions for this article.

1. The format of lines 49 and 50 in the material section is incorrect

2. Only the fly ash in Table 1 is mentioned in the article and no mineral powder is mentioned

3. National standards should be quoted, not stated in the article as currently

4. The temperature in line 65 has no units, and should the strength be consistent with the uniaxial compressive strength below?

5. Is the expansion rate of line 66 defined by the author himself? If not, please quote

6. The abbreviation in line 147 should be consistent in the full text table, and the format of this part needs to be adjusted

7. The author only conducted the sem test as a micro-analysis. In my opinion, the author should add some micro-tests (such as spectrum or thermogravimetry) to prove the experimental results and discussion mentioned above

6. PLOS authors have the option to publish the peer review history of their article (what does this mean?). If published, this will include your full peer review and any attached files.

Reviewer #1: No

Reviewer #2: No

---

## [Author Response · Author response to Decision Letter 0]

31 May 2020

Dear Prof./Dr. Kedsarin Pimraksa:

Thank you very much for considering our manuscript. All authors would like to take this opportunity to express our gratitude to you and the anonymous reviewers for the valuable suggestions on our paper (Paper no: PONE-D-20-00990, Paper title: Experimental study of high-flow and low-expansion backfill material).

According to the comments, we have carefully revised the manuscript and made it easy to be understood. The revised manuscript is attached to this email, and questions are all answered one by one in the enclosed files. We hope the revised manuscript can meet these requests and to be accepted.

I look forward to hearing from your decision as soon as possible.

Best regards.

Cheng Wang, Chun Wang, Zuqiang Xiong, Yuli Wang, Yafeng Han

 

To ACADEMIC EDITOR’s Evaluations:

Comment 1: In my point of view, this manuscript is required additionally an explanation on the experimental design in the experimental part. Please explain the role of each material used clearly and explain the amount basis of each material used in the experiments to strengthen scientific method.

Response: Thank you for your valuable and thoughtful comments. The investigated backfill material in this study is intended to be used in Gushuyuan coal mine located in Jincheng City, China. In the initial investigation phase, a numerical modelling was conducted to determine the reasonable backfill material strength. The numerical model is shown in Fig. 1. Variation curve of the roof subsidence under different backfill material strength is shown in Fig. 2. In Fig.2, the abscissa is the strength ratio between the backfill material and coal, and the ordinate is the roof subsidence. It is observed that the roof subsidence gradually stabilizes after increasing the strength ratio to 0.6-0.8. Thus, the reasonable strength ratio of backfill material used in Gushuyuan coal mine was determined as 0.6-0.8. 

Fig. 1 Numerical model

Fig. 2 Variation curve of the roof subsidence under different backfill material strength 

The fly ash has the potential pozzolanic activity. Some of the gels (e.g., C-S-H and C-A-H) can be generated after mixing fly ash with lime and water. Additionally, the wide source and low price of the fly ash are also its advantage. Thus, the fly ash was selected as the main component of the backfill materials. The performance of the backfill material (e.g. the strength and bleeding) may be improved by reducing the amount of the fly ash and simultaneously increasing other highly actives. Nevertheless, the cost of the backfill material at this moment will be a headache. On the other hand, the backfill material properties are poor when the amount of the fly ash is high and the amount of other highly actives is low. The performance of the backfill materials is balanced when the mass ratio of the fly ash is 80% based on the initial investigation. The strength of the backfill material is sufficient at this ratio. Additionally, the cost of the backfill material is also cheap. Thus, the mass ratio of the fly ash was fixed at 80% in this study. 

The mine powder, lime, and desulfurized gypsum were selected as the cementitious material. The sodium carbonate was used as the alkali-activator to improve the hydration activity of the fly ash. The aluminum powder is used as air-entraining agent. A series of experiments were designed to determine the amount of the cementitious material, the alkali-activator, and the air-entraining agent. We have added a short description to explain the role of each material. The amount basis of each material used in the experiments were also explained. Please see lines 46-47, 50-51,55-56，page 3; lines 57, page 4; 73-77, page 4. 

To Review #1’s Evaluations:

Comment 1: Table 1: Change loss by loss in ignition

Response: Thank you very much, we have changed loss by loss in ignition.

Comment 2: Line 65: degree? ºC?

Response: Thank you very much, Line 65: the temperature was 20±1 ºC.

Comment 3: Line 158: changed Yuan by dollar?

Response: Thank you very much. As you suggested that we have changed Yuan by dollar, the backfill cost was 11.2 dollar /m3.

Comment 4: Line 186: add gel at CSH (II)

Response: Thank you very much. According to your suggested that we have added gel at CSH (II).

Comment 5: The references are very local, they should be more international.

Response: Thank you very much. As you suggested that we have added more international references. And the revised portions are marked in this manuscript.

[2] Pokharel, M., Fall, M., 2011. Coupled thermochemical effects on the strength development of slag-paste backfill materials. J. Mater. Civ. Eng. 23 (5):511-525.

[3] Ayub, T., Shafiq, N., Khan, S.U., Nuruddin, M., 2013. Durability of concrete with different mineral admixtures: a review. Int. J. Civ. Struct. Constr. Archit. Eng. 7 (8), 265-276.

[7] Koohestani, B., Belem, T., Koubaa, A., Bussière, B., 2016. Experimental investigation into the compressive strength development of cemented paste backfill containing nanosilica. Cem. Concr. Compos. 72:180-189.

[11] Fall, M., Benzaazoua, M., 2005. Modeling the effect of sulphate on strength development of paste backfill and binder mixture optimization. Cem. Concr. Res. 35 (2):301-314.

[16] Sahmaran, M., Yaman, I.O., Tokyay, M., 2009. Transport and mechanical properties of selfconsolidating concrete with high volume fly ash. Cem. Concr. Compos. 31 (2):99-106.

Comment 6: English must be revised.

Response: Thank you very much to point out this critical problem. As you suggested, we have asked a high quality professional English editorial service to edit this paper, and the certificate is along with this revision.

To Review #2’s Evaluations:

Comment 1: The format of lines 49 and 50 in the material section is incorrect.

Response: Thank you very much. According to your suggested that we have revised the format of lines 49 and 50 in the material section.

Comment 2: Only the fly ash in Table 1 is mentioned in the article and no mineral powder is mentioned.

Response: Thank you very much. Mineral powder as an additive, we really do not describe it in detail. However, the mineral powder is mentioned in line 74 and Table 2.

Comment 3: National standards should be quoted, not stated in the article as currently

Response: Thank you very much to point out the problem. The slump flow of the slurry, compressive strength of the consolidated slurry, and expansion rate were determined in this manuscript, respectively. The slump flow test of the slurry was performed based on Appendix A in GB 50119-2003. The preparation, maintenance and determination of compressive strength of the specimen were performed based on the GB/T 50080-2016. The expansion rate of the consolidated slurry was defined as the ratio of the interface height differences before and after slurry hardening to the interface height value of the initial slurry. The determination thought of the expansion rate is consistent with the expansive ratio determination of expansive cement (JC/T 313-2009). According to your suggested that the national standards have been quoted, and the quoted part have been marked in the paper. Please see lines 71-73, page 4. Once again, thank you for your valuable reminder. 

Comment 4: The temperature in line 65 has no units, and should the strength be consistent with the uniaxial compressive strength below?

Response: Thank you very much to point out the problem. We have added the units, the temperature was 20±1 ºC. And the strength is consistent with the uniaxial compressive strength below.

Comment 5: Is the expansion rate of line 66 defined by the author himself? If not, please quote

Response: Thank you very much. The expansion rate of line 66 defined is quoted by other author, the reference is listed as follows:

D.J. De Souza, L.F.M. Sanchez, M.T. De Grazia. Evaluation of a direct shear test setup to quantify AAR-induced expansion and damage in concrete. Construction and Building Materials 2019, 9(6):1-10.

Comment 6: The abbreviation in line 147 should be consistent in the full text table, and the format of this part needs to be adjusted.

Response: Thank you for your valuable comments. The abbreviation have been consistent in the full text table, and the format of this part have been also adjusted. And the revised parts have been marked in the paper.

Comment 7: The author only conducted the sem test as a micro-analysis. In my opinion, the author should add some micro-tests (such as spectrum or thermogravimetry) to prove the experimental results and discussion mentioned above.

Response: Thank you for your valuable and thoughtful comments. As you suggest, if the paper add some micro-tests (such as spectrum or thermogravimetry), it can prove the experimental results and the discussion part. Before we submitted the paper, we intended to do this experiment, but due to the laboratory problems and our uncertainty about the micro experiment, we did not do it. We are very grateful for your valuable advice, which verified our idea, because of the 2019-nCoV, we are unable to complete the experiment. We are very grateful for your comments, which provided the basis for our later research.

---

## [Editor Report · Decision Letter 1]

30 Jun 2020

PONE-D-20-00990R1

Experimental study of high-flow and low-expansion backfill material

PLOS ONE

Dear Dr. WANG,

Thank you for submitting your manuscript to PLOS ONE. After careful consideration, we feel that it has merit but does not fully meet PLOS ONE’s publication criteria as it currently stands. Therefore, we invite you to submit a revised version of the manuscript that addresses the points raised during the review process.

1. Inconsistent word: Authors used mineral powder and mine powder in many places.

2. Missing of characterization explanation: Sample preparation for SEM analysis should be explained in the part of experiment and methods.

We look forward to receiving your revised manuscript.

Kind regards,

Kedsarin Pimraksa, PhD

Academic Editor

PLOS ONE

Additional Editor Comments (if provided):

1. Inconsistent word: Author used both mineral powder and mine powder in many places.

2. Missing of characterization explanation: Sample preparation for SEM analysis should be explained in experimental and methods.

---

## [Author Response · Author response to Decision Letter 1]

9 Jul 2020

Dear Prof./Dr. Kedsarin Pimraksa:

Thank you very much for considering our manuscript. All authors would like to take this opportunity to express our gratitude to you and the anonymous reviewers for the valuable suggestions on our paper (Paper no: PONE-D-20-00990R1, Paper title: Experimental study of high-flow and low-expansion backfill material).

According to the comments, we have carefully revised the manuscript and made it easy to be understood. The revised manuscript is attached to this email, and questions are all answered one by one in the enclosed files. We hope the revised manuscript can meet these requests and to be accepted.

I look forward to hearing from your decision as soon as possible.

Best regards.

Cheng Wang, Chun Wang, Zuqiang Xiong, Yuli Wang, Yafeng Han

 

To Academic Editor’s Evaluations:

Comment 1: Inconsistent word: Author used both mineral powder and mine powder in many places.

Response: Thank you very much. As you suggested that we have changed the inconsistent word. And the revised portions are marked in this manuscript.

Comment 2: Missing of characterization explanation: Sample preparation for SEM analysis should be explained in experimental and methods.

Response: Thank you very much. According to your suggested that we have added the characterization explanation, and the revised portions are marked in this manuscript.

The SEM sample was taken from the specimens in uniaxial compression test, the sample was made by 1cm*1cm. Then it was fixed on the metal plate with double-sided conductive adhesive, the metal plate was placed in a gold-spraying apparatus for gold spraying, drying and vacuuming, and finally placed in a SEM instrument for the experiment.

---

## [Editor Report · Decision Letter 2]

14 Jul 2020

Experimental study of high-flow and low-expansion backfill material

PONE-D-20-00990R2

Dear Dr. WANG,

We’re pleased to inform you that your manuscript has been judged scientifically suitable for publication and will be formally accepted for publication once it meets all outstanding technical requirements.

Kind regards,

Kedsarin Pimraksa, PhD

Academic Editor

PLOS ONE
---

## [Editor Report · Acceptance letter]

30 Jul 2020

PONE-D-20-00990R2 

Experimental study of high-flow and low-expansion backfill material 

Dear Dr. WANG:

I'm pleased to inform you that your manuscript has been deemed suitable for publication in PLOS ONE. Congratulations! Your manuscript is now with our production department. 

Kind regards, 

on behalf of

Dr. Kedsarin Pimraksa 

Academic Editor

PLOS ONE